# Risk of Recurrent Pregnancy Loss in the Ukrainian Population Using a Combined Effect of Genetic Variants: A Case-Control Study

**DOI:** 10.3390/genes12010064

**Published:** 2021-01-05

**Authors:** Eleni M. Loizidou, Anastasia Kucherenko, Pavlo Tatarskyy, Sergey Chernushyn, Ganna Livshyts, Roman Gulkovskyi, Iryna Vorobiova, Yurii Antipkin, Oleksandra Gorodna, Marika A. Kaakinen, Inga Prokopenko, Ludmila Livshits

**Affiliations:** 1Section of Genetics and Genomics, Department of Metabolism, Digestion and Reproduction, Imperial College London, London W12 0NN, UK; loizidou@fleming.gr (E.M.L.); m.kaakinen@surrey.ac.uk (M.A.K.); 2Institute of Molecular Biology and Genetics NAS, 03143 Kiev, Ukraine; a.m@gmail.com (A.K.); tatarskyy@yahoo.com (P.T.); atgc@ukr.net (S.C.); ganna.livshyts@gmail.com (G.L.); falko0grv@gmail.com (R.G.); algora@i.ua (O.G.); 3Institute of Paediatrics, Obstetrics and Gynaecology of the National Academy of Medical Sciences, 04050 Kiev, Ukraine; doctor.irina.v@gmail.com (I.V.); dzina7@ukr.net (Y.A.); 4Section of Statistical Multi-Omics, Department of Clinical & Experimental Medicine, School of Biosciences & Medicine, University of Surrey, Guildford GU2 7XH, UK; 5Institute of Biochemistry and Genetics, Ufa Federal Research Centre Russian Academy of Sciences, 119192 Ufa, Russia

**Keywords:** recurrent pregnancy loss, genetic risk score

## Abstract

We assessed the predictive ability of a combined genetic variant panel for the risk of recurrent pregnancy loss (RPL) through a case-control study. Our study sample was from Ukraine and included 114 cases with idiopathic RPL and 106 controls without any pregnancy losses/complications and with at least one healthy child. We genotyped variants within 12 genetic loci reflecting the main biological pathways involved in pregnancy maintenance: blood coagulation (*F2*, *F5*, *F7*, *GP1A*), hormonal regulation (*ESR1*, *ADRB2*), endometrium and placental function (*ENOS*, *ACE*), folate metabolism (*MTHFR*) and inflammatory response (*IL6*, *IL8*, *IL10*). We showed that a genetic risk score (GRS) calculated from the 12 variants was associated with an increased risk of RPL (odds ratio 1.56, 95% CI: 1.21, 2.04, *p =* 8.7 × 10^−4^). The receiver operator characteristic (ROC) analysis resulted in an area under the curve (AUC) of 0.64 (95% CI: 0.57, 0.72), indicating an improved ability of the GRS to classify women with and without RPL. Ιmplementation of the GRS approach can help define women at higher risk of complex multifactorial conditions such as RPL. Future well-powered genome-wide association studies will help in dissecting biological pathways previously unknown for RPL and further improve the identification of women with RPL susceptibility.

## 1. Introduction

The loss of two or more sequential pregnancies in the first trimester of gestation is defined as recurrent pregnancy loss (RPL) [1]. It affects nearly one in 20 women of reproductive age [2]. The vast majority of early PL (50–60%) are the consequence of chromosomal abnormalities including both numerical abnormalities and structural alterations [3,4]. In RPL etiology, endocrine, immunological, anatomical and other factors are proposed to play a leading role, with most of the remaining RPL cases being idiopathic [5,6]. For example, conditions such as chronic endometritis (CE) have been associated with RPL, and there are reliable diagnostic techniques, including hysteroscopy and CD138 immunohistochemical stain to identify CE in RPL cases [7,8]. Additionally, several infections, such as toxoplasma gondii, may lead to RPL [9,10]. In many cases, however, the cause of RPL is yet unknown. In such cases, a multifactorial nature is usually suggested, with genetic component viewed as an important factor [5,11,12].

Suggested mechanisms underlying RPL include alterations in blood coagulation, hormonal regulation, endometrium and placental function, folate metabolism and inflammatory response. Three hemostasis-related genes (*F2*, *F5*, *F7*) in the blood coagulation pathway are strong candidates for RPL through their associations with venous thromboembolism and thrombosis [13,14,15,16], hereditary thrombophilia [17], RPL [18] and recurrent miscarriages [19]. The *GP1A* gene, in turn, is an important player in platelet adhesion to collagen [20]. Within hormonal regulation, estrogens modulate multiple reproductive functions, including progesterone production and uteroplacental blood flow [21]. Variation within the estrogen receptor gene *ESR1* has been linked to endometriosis susceptibility [22] and maternal age at first birth [23]. The stress-induced adrenergic receptor (*ADRB2*) activation may in turn affect embryo-maternal interactions during implantation, resulting in pregnancy complications and miscarriage [24]. Further, variation at *ADRB2* has been associated with preterm delivery [25,26].

Expression levels of angiogenesis-related genes, such as the endothelial nitric oxide synthase (*ENOS*) and angiotensin I-converting enzyme (*ACE*), are used to measure abnormalities of placental vasculature in the chorionic villi of RPL patients. Variability in these genes may result in PL, pre-eclampsia, intrauterine fetal death and growth restriction [27]. Moreover, variability in *ACE* has been associated with RPL [28].

Variation within the 5,10-methylenetetrahydrofolate reductase (*MTHFR*) gene is associated with higher serum folate [29] and homocysteine [30,31] concentrations. Mild elevations in the latter are a risk factor for placental abruption, infarction and pre-eclampsia [32], and are also associated with an increased risk of RPL [33,34,35,36,37]. Studies on variation at *MTHFR* in relation to RPL have provided inconsistent findings [37,38].

Finally, imbalances in the homeostasis between the fetal and maternal immune system may lead to pregnancy failure [39]. Variation at the inflammatory gene *IL10* has been associated with early PL [40] and RPL [41], whereas the evidence for *IL6* and RPL is unclear [42]. MicroRNA studies for endometriosis have shown the potential role of IL-8 levels in the pathogenesis of endometriosis via stimulating endometrial stromal cell invasiveness [43].

## 2. Materials and Methods

We genotyped variants at/within the 12 above-mentioned genes in a Ukrainian sample of 114 cases and 106 healthy controls and aimed to evaluate the predictive ability of this combined gene set to the risk of RPL.

### 2.1. Study Sample

The REPLIK (Recurrent Pregnancy Loss in Kiev) study case group comprised 114 unrelated women with a mean age of 34.2 (SD 4.5) years and idiopathic RPL history undergoing observation in the State Institution “Institute of Pediatrics, Obstetrics and Gynecology of NAMS of Ukraine” and perinatal clinic “ISIDA”. All the women were of Ukrainian descent from across Ukraine. RPL diagnosis was determined in case of at least two consequent miscarriages in the first trimester (mean number of fetal losses 2.7, SD 0.9). The American Society for Reproductive Medicine defines RPL as two or more clinical pregnancy losses, not necessarily consecutive, documented by ultrasonography or histopathologic examination [44]. In order to ensure the idiopathic nature of RPL in the studied patients, the following enrolment criteria were set: the absence of a family history of birth defects; absence of the genital tract anatomic abnormalities, confirmed by ultrasonography or hysterosalpingography; a normal karyotype of both the studied individual as well as their partner, defined by GTG-banded chromosome analysis including GTG-banded metaphase plates with a minimum resolution of 400–450 bands per each sample. Moreover, blood tests for immunologic risk factors (anti-nuclear antibodies, anti-phospholipid antibodies, lupus anticoagulant), defects of thyroid function, diabetes mellitus, hyperprolactinemia and infections such as chlamydia were performed, and none of the individuals who tested positive were included in the study group. A control group comprised of 106 unrelated healthy women at a mean age of 26.2 (SD 3.0) years with no history of RPL or other pregnancy complications, no fetal losses, and having given birth to at least one naturally conceived child. Prior to the clinical examination and genotyping, all participants had given their informed consent. The study was approved by The Bioethical Committee of the Institute of Molecular Biology and Genetics of the National Academy of Sciences (NAS), Ukraine.

### 2.2. Blood Sample Collection, DNA Extraction and Genotyping Procedures

Venous blood samples from patients and control group individuals were collected into 4 mL vacutainer tubes containing EDTA. Genomic DNA was extracted from blood samples by the standard method using proteinase K with chloroform extraction. We used 13 genetic variants previously associated with RPL in European populations or those representing genetic networks of pathological processes leading to RPL (Table 1 provides information about the genotyped variants and the respective reference papers). The SNP variant located in the *IL10* gene (rs1800872) was excluded from the polygenic risk score calculation due to high LD (*r*^2^ = 0.26 in Europeans) with a nearby SNP included in the study (rs1800896). We had the same number of individuals genotyped for both SNPs within the *IL10* gene and hence, randomly chose the SNP rs1800896 to be included in the PRS. Genotyping for the selected polymorphic variants was performed by common variations of PCR-based assays as described previously (Table 1) with slight modifications.

We defined the genotypes for all individuals without any missing genetic data. The number of genotyped individuals varies per variant because genotyping of the 13 variants was done for the individuals depending on their time of enrolment into the project and availability of the reagents (Table 2 shows the numbers of cases and controls genotyped for each variant).

### 2.3. Association Analysis

We performed single-variant association analyses between each genotyped variant and RPL. We used logistic regression for single-variant analyses assuming a log-additive model of association, similar to standard assumptions in the genome-wide association study (GWAS). We reported estimates of odds ratios (ORs) along with their 95% confidence intervals (CIs).

### 2.4. Genetic Risk Score Calculation

We calculated the genetic risk score (GRS) using information from a set of 12 variants previously associated with RPL (Table 1). Among genotyped variants, there are 10 SNPs, an insertion/deletion (indel) rs1799752 at the *ACE* gene and a tandem repeat (VNTR) in intron 4 of the *ENOS* gene. The risk allele count for each SNP was weighted by its established effect size using previously published findings. The effect sizes were estimated from the ORs in the form of beta coefficients (log (OR)) for an association between GRS and RPL assuming an additive genetic model. The weighted GRS was corrected for genotypes unavailable for some individuals by multi plying the score with the total number of variants and then dividing by the number of genotyped variants per person, that is, the scores for people with less genotyped variants got more weight [59]. The effect estimates for the Intron-4 VNTR within *ENOS*, for the tandem repeat 14718 within *IL6* and for rs1800896 located at *IL10* were from multi-ethnic studies, discordant from our study’s ethnic descent. Therefore, we used the present study effect estimates. In addition, for the variants in *F7*, *GP1A*, *ADRB2* and *IL8* there were no published studies for their association with RPL. Hence, we used the effect sizes from our data as weights. As a sensitivity check we also calculated the GRS only using weights from our study. Finally, as an additional sensitivity check, we calculated an unweighted GRS and evaluated its effect on RPL.

### 2.5. Receiver Operating Characteristic Analysis

Next, we performed receiver operating characteristic (ROC) analysis to determine the predictive value of the estimated GRS in RPL. The efficacy of the GRS prediction is measured using the area under the curve (AUC), which is the statistic calculated on the observed case scale. The statistical analyses were conducted using the statistical software pROC package in R along with other functions [60].

## 3. Results

### 3.1. Association Analysis

We tested the 13 genetic variants for association with RPL in the REPLIK study from Ukraine. Within this variant set only rs1800896 at *IL10* gene was nominally associated with RPL (OR (95% CI): 1.60 (1.07, 2.42), *p* = 0.025, Table 2). Although the other variants did not reach nominal significance, six of the eight with an effect estimate available from the literature were in the same direction of the effect in our study as in the published ones (Table 1 and Table 2). The sample sizes varied greatly for the genotyped SNPs, whereas the GRS accounted for the variable number of genotyped SNPs per individual. The association between RPL and the GRS, based on published effect estimates, revealed a statistically significant association (*p* = 8.7 × 10^−4^) in the REPLIK study. The combined effect of all tested variants resulted in 1.56-fold (95% CI (1.21, 2.04)) higher odds of RPL between cases and controls. The sensitivity analysis using the GRS with weights from our study effect estimates resulted in an OR of 1.83 (95% CI (1.34, 2.57), *p* = 3.0 × 10^−4^). The unweighted GRS was also associated with an increased risk of RPL with an OR of 1.16 (95% CI (1.05, 1.29), *p* = 2.0 × 10^−3^).

### 3.2. Receiver Operator Characteristic Analysis

The ROC analysis showed an area under the curve (AUC) of 0.64 (95% CI (0.57, 0.72)). This indicates a moderate to high ability of the GRS to correctly classify women with and without RPL. The sensitivity of 72% at the best discriminating point implies that the GRS can effectively identify women having experienced pregnancy losses (Figure 1). The AUCs for the GRS using our study effect estimates as weights and for the unweighted GRS were 0.64 (95% CI (0.57, 0.71)) and 0.62 (95% CI (0.55, 0.70)), respectively.

## 4. Discussion

In this study we evaluated the combined effect of a set of genetic variants on RPL risk through GRS implemented in a case-control REPLIK study from Ukraine. We showed that a carefully chosen genetic variant set is already useful for achieving predictive ability when implementing weighted GRSs.

A clear strength of our study was the well-defined RPL phenotype allowing us to detect differences between cases and controls in their genetic risk. The identification of RPL cases is laborious and expensive, hindering the setup of a well-powered GWAS for this outcome. Many studies, including the UK Biobank consisting of 500,000 individuals, have collected data on self-reported miscarriages and the number of spontaneous miscarriages. GWAS on the UK Biobank data [61] on such surrogate phenotypes for RPL have not resulted in any common variants associated with RPL at genome-wide significance, plausibly reflecting an inaccurate re-call and complex genetic susceptibility to RPL.

Another strength of our study was the careful selection of the gene panel based on the hypothesized biological pathways. We had previous evidence for the association with RPL for some of the variants (within *F2*, *F5*, *ACE*, *IL10*) used [18,28,40], whereas others (within *F7*, *GP1A*, *ESR1*, *ADRB2*, *ENOS*, *MTHFR*, *IL6*, *IL8*) were chosen for their hypothesized biological mechanisms [19,20,25,26,38,42,43,48,51]. For the variants where we found an opposite direction of effect with the published results, the ancestry of the studied population differed from ours. That is, we investigated RPL in Ukrainian women whereas the results from previous studies also reported associations for US, Asian and African women [28,38,41,42,51]. Overall, by combining the effects of our collected variants, we could already predict the risk of RPL in our cohort. This is particularly important since, for women with unexplained RPL, there is currently not enough evidence that justifies the use of in-vitro fertilization [62]. Our study adds information about the genetic causes of RPL and could potentially assist in the design of improved treatments.

We achieved an AUC of 0.64 (95% CI 0.57–0.72) with a set of 12 SNPs only. A recent study combining millions of SNPs into genome-wide polygenic scores for several complex diseases achieved AUCs of similar strength. For example, an AUC of 0.63 was reported for inflammatory bowel disease from a GRS consisting of 6.9 million SNPs [63]. Taken together, our results are important considering that RPL is a laborious and expensive phenotype to collect. Since the start of large-scale GWAS, RPL has lacked novel insights establishing its underlying genetic mechanisms with no major publications probably due to clinical requirements to the phenotype definition. However, our investigation suggests that even a small number of SNPs in appropriately defined cases and controls can be used for predictive purposes.

It has been acknowledged that the discrete genetic variants for RPL have relatively low sensitivity and specificity [64]. Each individual case of idiopathic RPL usually cannot be explained by one risk factor and should be treated as a multifactorial condition [11]. Indeed, GWAS for complex traits has shown that individual genetic variants usually provide a relatively modest contribution to the trait variability in terms of their per-allele effect size, typically in the per-allele effects being within the range of 5–10% increase in risk in relation to that of risk estimated in the general population [65], and hence require large sample sizes for detection. Therefore, it may be more effective to evaluate the risk of RPL using a panel of population-specific low-effect genetic markers, representing distinct physiological gene networks.

A limitation of our study was the relatively small sample size, and hence, we could not confirm the associations with RPL for a majority of the variants per se with our available sample. This could, however, also reflect the difficulty in establishing genetic associations with the complex condition of RPL. The sample sizes varied for the variants due to our genotyping strategy. For the *IL10* variant, having the whole sample genotyped, we demonstrated its association with RPL with an effect estimate similar to a previous study [40]. The ideal approach would have been to genotype each variant in the same set of individuals or to impute missing genotypes. However, genotype imputation relies on the knowledge of the regional linkage disequilibrium structure and hence access to genome-wide data. To account for the missing data in the GRS analysis we applied an approach suggested by Belsky et al. [59] to weight the risk score by the number of genotyped variants per person.

## 5. Conclusions

With the careful selection of the DNA variant set and the implementation of methods such as the GRS, we can predict susceptibility to complex multifactorial conditions such as RPL. Future well-powered studies, especially GWAS, adding to the knowledge of biological pathways previously unknown for RPL, will significantly improve the prediction and identification of women at risk for RPL.

## Figures and Tables

**Figure 1 genes-12-00064-f001:**
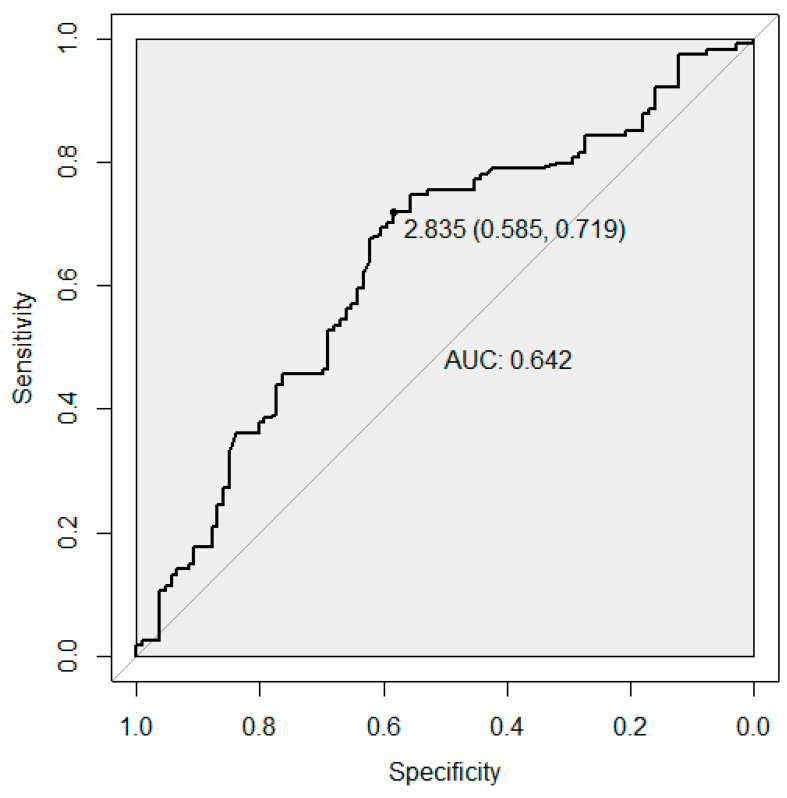
Receiver operator characteristic (ROC) curve for the predictive ability of the genetic risk score (GRS) for recurrent pregnancy loss. The best predictive point is shown with the ideal cut-off for the GRS and with estimates for specificity and sensitivity at that point. Abbreviation: AUC, area under the curve.

**Table 1 genes-12-00064-t001:** Genetic variants used to assess their effect on the risk of RPL in the REPLIK study.

Locus Name	SNP rsID	EA/NEA	EAF (1000 G *)	OR (95% CI)	*p*-Value	Case/Control Sample Size	Outcome	Reference	Genotyping Method	Reference
Blood coagulation										
*F2*	rs1799963	A/G	0.008	2.00 (1.00, 4.00)	<0.03	342/123	RPL	[18]	RFLP	[45]
*F5*	rs6025 ^$^	T/C	0.012	2.5 (1.80, 3.40)	<10^−3^	342/123	RPL	[18]	RFLP	[45]
*F7*	rs6046	G/A	0.89	-	-	-	-	-	RFLP	[46]
*GP1A*	rs1126643	T/C	0.40	-	-	-	-	-	RFLP	[47]
Hormonal regulation										
*ESR1*	rs2234693 ^^^	T/C	0.58	1.10 (0.57, 2.13)	>0.05	350/646	RPL	[48]	RFLP	[49]
*ADRB2*	rs1042714	G/C	0.41	-	-	-	-	-	RFLP	[50]
Endometrium and placental function										
*ENOS*	Intron-4 ^ǂ^ VNTR ^^^	B/A	-	1.005 (0.74, 1.37)	>0.05	410/357	RPL	[51]	Allele-specific PCR	[52]
*ACE*	rs1799752 ^†,$^	D/I	-	2.06 (1.46, 2.91)	NA	740/329	RPL	[28]	Allele-specific PCR	[53]
Folate metabolism										
*MTHFR*	rs1801133 ^$^	T/C	0.36	1.25 (0.93, 1.67)	0.14	1830/3037	RPL	[38]	RFLP	[54]
Inflammatory response										
*IL6*	14718 ^$^	G/C	-	1.214 (0.88, 1.67)	0.24	230/188	RPL	[42]	RFLP	[55]
*IL8*	rs2227306	C/T	0.61	-	-	-	-	-	RFLP	[56]
*IL10*	rs1800896 ^^^	G/A	0.45	1.27 (0.95, 1.70)	>0.05	635/571	RPL	[41]	RFLP	[57]
*IL10*	rs1800872	T/G	0.76	3.01 (1.92, 4.72)	<10^−^^4^	342/123	Early PL	[40]	RFLP	[58]

Legend: ^†^—Indel; ^ǂ^—Tandem repeat; EA/NEA—Effect allele/Non-effect allele; * EAF (1000 G)—1000 Genomes project effect allele frequency in Europeans; ^^^—Non-European population; ^$^—Multi-ethnic populations.

**Table 2 genes-12-00064-t002:** Associations between 12 previously reported genetic variants and RPL in the REPLIK study.

Locus Name	Chr: Position	Variant ID	EA/NEA	EAF Cases/Controls	N Cases/N Controls	EAE	OR (95% CI)	*p*-Value
*F2*	11:46761055	rs1799963	A/G	0.018/0.011	110/46	0.49	1.64 (0.23, 32.48)	0.66
*F5*	1:169519049	rs6025	A/G	0.018/0.022	114/46	−0.22	0.80 (0.15, 5.92)	0.80
*F7*	13:113773159	rs6046	G/A	0.89/0.85	75/46	0.34	1.40 (0.65, 3.04)	0.38
*GP1A*	5:52347369	rs1126643	C/T	0.63/0.54	81/46	0.35	1.43 (0.85, 2.44)	0.18
*ESR1*	6:152163335	rs2234693	T/C	0.49/0.55	110/106	−0.18	0.84 (0.57, 1.23)	0.37
*ADRB2*	5:148206473	rs1042714	C/G	0.60/0.51	81/46	0.38	1.47 (0.86, 2.56)	0.16
*ENOS*	15:35147732-35262040	Intron-4 ^ǂ^ VNTR	B/A	0.83/0.80	102/46	0.15	1.16 (0.63, 2.08)	0.63
*ACE*	17:61565890	rs1799752 ^†^	D/I	0.52/0.53	100/46	−0.048	0.95 (0.59, 1.54)	0.85
*MTHFR*	1:11856378	rs1801133	T/C	0.30/0.24	114/46	0.33	1.40 (0.80, 2.51)	0.25
*IL6*	7:22766840	14718	G/C	0.59/0.56	106/106	0.11	1.10 (0.77, 1.64)	0.57
*IL8*	4:74607055	rs2227306	T/C	0.40/0.37	114/106	0.11	1.10 (0.76, 1.64)	0.60
*IL10*	1:206946897	rs1800896	A/G	0.59/0.49	114/106	0.47	1.60 (1.07, 2.42)	0.025
GRS					114/106		1.56 (1.21, 2.04)	8.7 × 10^−4^

Legend: ^†^ Indel; ^ǂ^ Tandem repeat; EA/NEA: Risk allele/Alternate allele; EAF: Effect allele frequency; EAE: Estimated allelic effect (beta); GRS, genetic risk score.

## Data Availability

The data presented in this study are available on request from the corresponding author. The data are not publicly available due to privacy restrictions.

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
