# Peer review of "Risk of Recurrent Pregnancy Loss in the Ukrainian Population Using a Combined Effect of Genetic Variants: A Case-Control Study"

_genes, 2021, doi:10.3390/genes12010064_

Round 1

Reviewer 1 Report

I read with great interest the Manuscript titled “Risk of recurrent pregnancy loss in the Ukrainian population using a combined effect of genetic variants: a case-control study” (genes-1015298), which falls within the aim of Genes.    

In my honest opinion, the topic is interesting enough to attract the readers’ attention. The methodology is accurate and conclusions are supported by the data analysis. Nevertheless, the authors should clarify some points and improve the discussion by citing relevant and novel key articles about the topic.

Authors should consider the following recommendations:

  • The manuscript should be further revised by a native English speaker.
  • What are the actual clinical implications of this study? it is important to report the results obtained by the authors in the context of clinical practice and to adequately highlight what contribution this study adds to the literature already existing on the topic and to future study perspectives.
  • Different studies demonstrated the relation between chronic endometritis and recurrent pregnancy loss. It would be interesting to discuss, even briefly, this point referring to: PMID: 30851430; PMID: 26207958.
  • Some infections can be also a cause of recurrent pregnancy loss. The Authors can discuss this point by referring to: PMID: 32013639; PMID: 32833877.

Reviewer 2 Report

I found this study to be interesting, generally well-written, and the results are striking.  Additionally, the definition of RPL with rigorous exclusion/inclusion criteria make this study more robust than other genetic association studies of RPL.

However, I have some serious concerns about the association analyses as shown in Table 2, and about how missing data was dealt with.

In table 2, the effect allele frequencies are unreasonable for several SNPs. For the first SNP, in the F2 locus, the EAF for cases was only 0.011 and for controls was 0.72. This vast difference in EAF in cases and controls is simply not plausible. The EAF in 1000G for that SNP is 0.008, indicating that the reported EAF for cases of 0.011 is likely correct but 0.72 for controls is not.  The SNPs in F5, F7, ENOS, ACE, and MTHFR also had exceedingly large differences in the EAF between cases and controls, indicating possible errors either in genotyping, calculating, or in reporting.

Additionally, the odds of carrying the effect allele in cases divided by the odds of carrying the effect allele in controls should ~approximate the odds ratio, or at least be in the same direction. (I do understand that the number of effect alleles carried by an individual also matters in the log-additive model).  Analyses were unadjusted for any confounders, or at least none were mentioned in the methods. For the first SNP in F2, the cases were far less likely to carry the effect allele than controls (EAFs of 0.011 vs. 0.72) Yet somehow the odds ratio is 1.64, implying that cases were more likely to carry the effect allele. This is not plausible. The odds ratio should be well below 1.0, since the effect allele was far less frequent in cases than controls.  Several other SNPs in Table 2 have the same problem, where the EAFs in cases vs. controls are not consistent with the direction of the odds ratio (ESR1, ACE, MTHFR, IL8).  

For ACE, the reported EAF among the controls (0.0023) is not mathematically possible. Since there are 46 controls for ACE, there are 92 indel alleles. This means that if there is even a single effect allele among the controls, the EAF would be 1/92 or 0.0108. 

My second concern is with the weighting method for the risk score in the presence of missing data.  If an individual was missing a few SNPs, then the SNPs that they had were simply over-weighted accordingly to make up for the missing SNPs. This may cause bias if those SNPs had a large effect.  The authors should use another method such as multiple imputation to fill in the missing SNPs. 
